# Nontuberculous Mycobacteria, Mucociliary Clearance, and Bronchiectasis

**DOI:** 10.3390/microorganisms12040665

**Published:** 2024-03-27

**Authors:** Miriam Retuerto-Guerrero, Ramiro López-Medrano, Elizabeth de Freitas-González, Octavio Miguel Rivero-Lezcano

**Affiliations:** 1Servicio de Reumatología, Complejo Asistencial Universitario de León, Gerencia Regional de Salud de Castilla y León (SACYL), Altos de Nava, s/n, 24071 León, Spain; mretuertog@saludcastillayleon.es; 2Servicio de Microbiología Clínica, Complejo Asistencial Universitario de León, Gerencia Regional de Salud de Castilla y León (SACYL), Altos de Nava, s/n, 24071 León, Spain; rzlopez@saludcastillayleon.es; 3Servicio de Neumología, Complejo Asistencial Universitario de León, Gerencia Regional de Salud de Castilla y León (SACYL), Altos de Nava, s/n, 24071 León, Spain; emfreitas@saludcastillayleon.es; 4Unidad de Investigación, Complejo Asistencial Universitario de León, Gerencia Regional de Salud de Castilla y León (SACYL), Altos de Nava, s/n, 24071 León, Spain; 5Institute of Biomedical Research of Salamanca (IBSAL), 37007 Salamanca, Spain; 6Institute of Biomedicine (IBIOMED), University of León, 24071 León, Spain

**Keywords:** ciliopathies, Lady Windermere syndrome, connective tissue diseases, *Pseudomonas aeruginosa*, biofilm, primary ciliary dyskinesia, bronchiectasis, autoimmune diseases, *Mycobacterium abscessus*

## Abstract

Nontuberculous mycobacteria (NTM) are environmental and ubiquitous, but only a few species are associated with disease, often presented as nodular/bronchiectatic or cavitary pulmonary forms. Bronchiectasis, airways dilatations characterized by chronic productive cough, is the main presentation of NTM pulmonary disease. The current Cole’s vicious circle model for bronchiectasis proposes that it progresses from a damaging insult, such as pneumonia, that affects the respiratory epithelium and compromises mucociliary clearance mechanisms, allowing microorganisms to colonize the airways. An important bronchiectasis risk factor is primary ciliary dyskinesia, but other ciliopathies, such as those associated with connective tissue diseases, also seem to facilitate bronchiectasis, as may occur in Lady Windermere syndrome, caused by *M. avium* infection. Inhaled NTM may become part of the lung microbiome. If the dose is too large, they may grow excessively as a biofilm and lead to disease. The incidence of NTM pulmonary disease has increased in the last two decades, which may have influenced the parallel increase in bronchiectasis incidence. We propose that ciliary dyskinesia is the main promoter of bronchiectasis, and that the bacteria most frequently involved are NTM. Restoration of ciliary function and impairment of mycobacterial biofilm formation may provide effective therapeutic alternatives to antibiotics.

## 1. Introduction

Bronchiectasis are dilatations of the bronchi characterized by excessive sputum production and increased risk of infections. Currently regarded as irreversible, some reports have lately speculated about potential improvements after chemotherapy [1]. Once considered an orphan disease, it has received an increasing focus in the past two decades [2]. It is believed to be a consequence of insufficient clearance of mucus from the airways, which facilitates infection and inflammation, worsening bronchiectasis in a vicious circle first described by Cole [3]. It is a very complex disorder, with multiple etiologies including, as the most important, post-infection bronchiectasis, mainly post-tuberculosis [4]. Increasing numbers of cystic fibrosis and non-cystic fibrosis bronchiectasis patients are affected by nontuberculous mycobacteria (NTM) and are more likely to have NTM infection compared to those without bronchiectasis [5]. In this review, we analyzed the known and potential relationships between nontuberculous mycobacteria infection, impaired mucociliary clearance, and the development of bronchiectasis.

## 2. Nontuberculous Mycobacteria

The relationship between bronchiectasis and mycobacteria is strong. There are two radiographic presentations of nontuberculous mycobacterial lung disease, fibrocavitary and nodular/bronchiectatic [6], although the fibrocavitary manifestation also usually appears as bronchiectasis [7]. Mycobacteria are in the *Mycobacterium* genus and a generally accepted classification divides them into three groups. The first one has one member, *Mycobacterium leprae*; the second is the *M. tuberculosis* complex, with several species including *M. tuberculosis* and *M. bovis*; and the third is the nontuberculous mycobacteria (NTM) that encompasses the rest of the mycobacteria. 

Unlike *M. tuberculosis*, considered an obligate pathogen, NTM are generally regarded as environmental opportunistic microorganisms. This is a narrow vision of the mycobacteria because *M. tuberculosis* survives in soil and water ecosystems [8], which are a potential sources of infection, and human-to-human transmission of NTM has been suggested for *M. abscessus* [9]. Twenty-five years ago, NTM were considered unimportant in cystic fibrosis patients [10], but the incidence and number of deaths from mycobacterial disease have increased globally in the last few decades [11]. The pathogenicity of the NTM species varies widely. The clinical relevance of different species, defined as the percentage of patients with isolates that meet the American Thoracic Society and the Infectious Diseases Society of America (ATS/IDSA) diagnostic criteria [6], has been estimated to range from 0% (*M. terrae*) to 88% (*M. intracellulare*) [12]. The *M. avium* complex, which includes *M. intracellulare* and *M. avium*, are the most common clinically associated NTM, followed by the rapidly growing *M. abscessus* complex [11,12]. Nevertheless, it is possible that all mycobacteria could cause lesions under special circumstances. Clinical cases involving rare mycobacteria continuously appear in the scientific literature [13]. The reasons that explain the difference in pathogenicity between the *M. tuberculosis* complex and NTM, or among different NTM species, have not been well clarified. Acknowledged virulence factors in *M. tuberculosis* are very diverse, such as cell envelope proteins, protein kinases, or proteases [14]. The genetic analysis of some factors among the different mycobacteria, including *M. tuberculosis*, reveal that each of them are shared by some or many species [15]. 

Although the clinical evolution of tuberculosis and other mycobacteriosis has been intensively studied, the initial events that take place when the microorganisms are inhaled are little known. The general assumption is that bacteria that are not trapped in the mucus and reach the alveolar space will be phagocytosed by alveolar macrophages to eliminate them [16]. However, *M. tuberculosis* adapts in alveolar epithelial cells where the bacilli are highly replicative and invasive [17]. Studies using ex vivo tissue culture models confirm that different species of mycobacteria infect non-phagocytic cells such as pneumocytes [18]. It is even conceivable that low doses of inhaled mycobacteria, whose growth is inhibited by innate humoral defense mechanisms such as defensins [19] or secretory leukocyte protease inhibitor [20], remain free in the airway lumen without eliciting an inflammatory response, becoming part of the normal lung microbiota. Because NTM are environmental and ubiquitous, it is surprising that they are not more frequently associated with respiratory infections, suggesting that their interaction with the host is usually harmless. In fact, they may have a beneficial effect, as has been shown for the immunomodulatory activity of several NTM such as *M. indicus pranii* [21], *M. brumae* [22], or *M. vaccae* [23].

In contrast, large doses may facilitate the induction of biofilm formation by quorum sensing. In chronic respiratory infections, biofilms are usually present in the form of aggregates that may be non-surface attached [24]. Few reports document the in vivo formation of biofilm in mycobacteria, and most of them refer to *M. tuberculosis*. Thus, the biofilm appears as extracellular bacteria in secondary lesions of infected guinea-pigs, and as air-liquid pellicles within lung cavities previously formed from mycobacterial caseating granulomas [25]. Regarding NTM, *M. abscessus* biofilms have been shown to be attached to alveolar walls, but also as aggregates in sputum [26]. *M. abscessus* biofilms have also been observed in the wall of a lung cavity [27]. The growth as a biofilm confers advantages to mycobacteria. Mycobacterial populations are heterogeneous, a consequence of asymmetrical cell division that distribute cell wall components unevenly, resulting in variable responses to antibiotics and host immunity. There is a division of labor between different subpopulations, including sharing of siderophores, activation of dormant cells by resuscitation-promoting factors, sharing of biofilm with non-biofilm populations, or cell-interactions that facilitate macrophage killing [28]. Biofilms increases drug tolerance and facilitates immune system evasion [25]. Although the development of mycobacterial biofilm is expected in bronchiectasis, no reports have been published, probably due to a scarcity of studies.

There is little information about the fate of mycobacteria after a successful immune response or antibiotherapy. These situations stress the bacilli and some of them may become persisters, drug tolerant sub-populations that endure antibiotics through non-heritable mechanisms such as metabolic slowdown, transcriptional and translational responses to stress, or efflux pumps [29]. Stress may also induce the development of L-forms, a cell wall deficiency state of bacteria that cannot be acid-fast stained and is considered a tool for survival [30]. In all these circumstances, no clinical signs are evident, and their isolation at the microbiological laboratory will not be feasible. 

## 3. Airway Mucociliary Clearance

Breathing exposes a host to a large number of particles, allergens, debris and toxins, that are inhaled from the environment. Many of these particles are microorganisms, and a few of them are pathogenic. Nevertheless, healthy persons rarely suffer from respiratory infections. The main pulmonary defense responsible for this extraordinary feat is the mucociliary clearance (MCC) mechanism. It is based on the trapping of particles in a mucous layer on the airway epithelial surface that is cleared from the lung by cilia on the luminal side of ciliated cells [31].

Mucociliary epithelia line the airways and contain secretory cells that provide mucus, other molecules, and multiciliary cells (Figure 1A). There are several secretory cell types, including pulmonary neuroendocrine cells that are key communicators between the immune and nervous systems by secreting active molecules The main mucus secretory cells are the goblet cells, which together with ciliated cells, facilitate mucociliary clearance [32,33]. Club cells, previously known as Clara cells, are precursors of both ciliated and goblet cells. In the alveoli epithelial cells, Type I mediates gas exchange, and Type II produces surfactant lipids and proteins that diminish surface tension. Together with epithelial cells, alveolar macrophages regulate the size of surfactant pools and remove microbial pathogens [33]. Normal airway mucus is a hydrogel comprised of approximately 97.5% water, 0.9% salt, 1.1% globular proteins, and 0.5% mucin polymers. In the airways, the most important polymers are MUC5AC and MUC5B. Glycan side chains bind large amounts of liquid hundreds of times their weight, which affects the critical viscous and elastic properties of mucus that makes it appropriate for ciliary clearance [34,35].

The remaining cell type that line the airways is the multiciliate epithelial cell, which transports the mucus layer from the distal to the proximal airways. Cilia are microtubule-based organelles present in most cells (Figure 1B). Most cell types have a single cilium (primary cilium), while some cells are multiciliated with bundles of 200–300 cilia. The core structure is the axoneme, with nine outer doublet microtubules (9 + 0). Some cilia have an additional central pair (9 + 2). The outer doublet consists of a complete tubule, the A tubule, with 13 tubulin subunits, and a partial tubule, the B tubule, with 11 subunits. These doublets are connected to each other by the nexin-dynein regulatory complex. The radial spokes connect the outer with the central doublets, and participates in both the mechanical stability of the axoneme and the regulation of ciliary activity. Attached to the A tubule are protein complexes, the inner (IDA) and the outer (ODA) dynein arms. Studies on the flagellated green alga *Chlamydomonas* suggest that the ODA are responsible for adjusting ciliary beat frequency, whereas the IDA are responsible for bend formation and waveform. Genomic and proteomic studies have shown that the cilium contains approximately 600 proteins [31].

Cilia are very complex organelles that play very different roles in other organs. Motile multiciliated cells with the 9 + 2 structure cilia similar to epithelial lung cells are the ependymal cells of the central nervous system that circulate cerebrospinal fluid within the brain and spinal cord. The motile monocilia (a single cilium per cell) is characteristic of sperm cells, in which the cilium is the denominated flagellum. Immotile cilia, also called sensory or primary cilia, are generally short, have the 9 + 0 structure, and lack dynein motility components, but are specialized to sense fluid flow, light, odorants, or signaling molecules [36]. This pattern of 9 + 2 motile and 9 + 0 immotile cilia is not always kept. An important 9 + 0 motile monocilium, with dynein arms, is present in the embryo’s left–right organizer organ of vertebrates, known as nodal cells in mice. Mutations in mice that paralyze these cilia were found to alter left–right asymmetry [37]. All these types of cilia share many components, but also have specific structures that confer specialized roles to them.

## 4. Bronchiectasis

The Cole’s vicious circle hypothesis proposes that bronchiectasis progresses from either a damaging insult, such as severe infectious pneumonia, or an underlying disease, such as cystic fibrosis, that affects the respiratory epithelium and compromises mucociliary clearance mechanisms, giving microorganisms the opportunity to colonize the airways [3] (Figure 1D).

### 4.1. Ciliary Dyskinesia

An adequate ciliary clearance requires (1) specific numbers of multiciliated cells per tissue area and numbers of cilia per cell; (2) each cilium must possess the correct ultrastructure (Figure 1B), and beat in the right direction at the optimum frequency and with the appropriate waveform (Figure 1C); and (3) cilia must interact with mucin macromolecules [38]. One or several of these properties are not fulfilled in ciliary dyskinesia, one of the underlying diseases and recognized risk factors for bronchiectasis [4]. Ciliary dyskinesia has been divided into primary (genetic disorders, inherited) and secondary (acquired). Primary ciliary dyskinesia (PCD) is genetically heterogeneous, with more than 50 associated genes to date [39]. Acquired or secondary ciliary dyskinesia are a consequence of a variety of agents’ effects that damage the respiratory epithelium, infection being the most important in bronchiectasis. The Swyer-James syndrome, a known risk factor for bronchiectasis, is thought to be a consequence of post-infectious bronchiolitis [40], and has been associated with mycobacteria infection and decreased ciliary function [41]. Some bacterial pathogens, such as *Pseudomonas aeruginosa* or *Mycoplasma pneumonia*, specifically target ciliated cells for adherence. Viruses are frequently involved, and HIV infection has been shown to reduce mucociliary clearance. Particularly relevant is cigarette smoking and passive smoking, which induce shorter cilia, patches of atypical nuclei, and missing cilia [42].

It is important to be aware of the difficulties of ciliary dyskinesia diagnosis, based on several procedures. (1) High-speed video-microscopy allows the analysis of ciliary beat frequency and the pattern of the respiratory epithelium obtained by nasal brush biopsy. Findings in PCD span from grossly abnormal to very subtle, and could even be impossible to detect [43]. For example, *DNAH11* variants present with hyperkinetic and stiff beat pattern, whereas *DRC4/GAS8* variants present with a rotational pattern [39]. (2) Transmission electron microscopy analysis, which reliably detect ODA, combined ODA/IDA or tubular disorganization defects, fails to recognize subtle abnormalities of ciliary ultrastructure present in 15–30% of PCD variants. (3) Immunofluorescence analysis uses fluorescently labeled antibodies against key ciliary proteins, which are visualized by microscopy [43], but depends on the availability of the appropriate labeled antibodies. (4) Measurement of nasal nitric oxide, found to be much lower in subjects with PCD [44], can also be reduced in other conditions such as recent viral infection, and some patients with PCD produce normal nitric oxide levels [43]. (5) Genetic testing is based on the detection of mutations in genes already related to the disease. It has been observed that motile ciliopathies are underdiagnosed in patients with bronchiectasis [45]. An intrinsic problem with this approach is that newly discovered genetic variants are still not included in commercial panels [43]. Furthermore, secondary ciliary dyskinesia will not be identified by genotyping. Therefore, ciliary deficiencies may be undetected by current diagnosis algorithms.

### 4.2. Concurrent Ciliopathies and Bronchiectasis

Specific additional clinical manifestations usually accompany bronchiectasis. For example, Kartagener’s syndrome, a subgroup of primary ciliary dyskinesia characterized by the triad bronchiectasis, sinusitis and situs inversus, is frequently associated with heterotaxy (situs ambiguos or organ laterality defects other than situs inversus totalis) and congenital cardiovascular disease [46]. Their concurrence may indicate a common underlying hereditary disorder, being the ciliopathies’ straightforward candidates. A connection has been established between heterotaxy/situs inversus and cilia [47,48]. A transient embryonic structure, termed “node” in the mouse, consists of epithelial cells that project a single cilium into a pit-like cavity filled with fluid. Cells in the pit have motile cilia, whereas surrounding “crown cells” have mainly immotile cilia. Motile cilia beat with a unidirectional vertical motion that results In a leftward fluid flow, inducing bending of the mechanosensory immotile primary cilia in the “crown cells”, which determine laterality. This embryonic structure is now known as the left–right organizer [48]. There is also a strong link between cilia and heart structural organization [49]. Several of the mutated genes in patients with primary ciliary dyskinesia are also associated with left–right patterning of the heart or with the possibility of randomization of left–right asymmetry during embryogenesis [50]. Therefore, there is some overlapping in the mutations of genes that affect both cells with primary cilia and multiciliated cells. Advances in sequencing technologies allow the constant discovery of genes that influence both types of cilia. Examples of genes initially characterized in primary ciliary dyskinesia but were later found in other ciliary functions are *CCDC114*, involved in sensorineural deafness and renal disease [51]; *CFAP52* [52] and *CCDC65* [53], which play a role in the male germ cell flagellum; or *DYX1C1* and other cilia genes in neuronal differentiation [54]. Nevertheless, the association is not perfect, because for example, no laterality defects have been reported for numerous variants of PCD [55,56]. The reason is that some genes characteristic of 9 + 2 cilia mutated in PCD, such as those encoding proteins of the radial spoke (Figure 1B), and are not expressed in the 9 + 0 cilia responsible for laterality [56]. The importance of ciliary dysfunctions and idiopathic bronchiectasis was exposed by the finding that patients had a greater proportion of cilia in the nasal respiratory mucosa with any ultrastructural microtubular defects or compound cilia, compared to healthy individuals [57].

The number of ciliopathies and candidate ciliopathy-associated genes is increasing, affecting both motile and non-motile (primary) cilia. In ciliopathies other than PCD, pulmonary manifestations and, particularly, bronchiectasis, have been described. A first example is the Bardet–Biedl syndrome, in which primary cilia are affected. In murine models, it was found that the motile cilia function was also impaired by the knock-out of the genes involved in the syndrome [58]. A subsequent study in patients found significant ciliary depletion and goblet cell hyperplasia [59], and a recent report documents bronchiectasias in a Bardet–Biedl syndrome patient [60]. The association between retinitis pigmentosa and PCD, usually complicated with bronchiectasis, is well documented in the scientific literature [61]. A significant relationship between polycystic kidney disease and bronchiectasis have been observed, and one of the genes mutated in the ciliopathy, polycistin-1, is expressed in motile cilia in the airway epithelial cells [62,63]. Bronchiectasis has also developed in a patient with Simpson–Golabi–Behmel syndrome [64], which has been related to mutations in *OFD1*; however, it is not certain that this disease is a ciliopathy [65]. Leber congenital amaurosis, produced by mutations in *CEP290*, also affects respiratory cilia, causing rarefaction of ciliated cells and short cilia [66]. The ciliopathy nephronophthisis has been associated with a mutation in *INVS/NPHP2* that also affects motile cilia [67]. Mutations in all these genes that encode centriolar or ciliary proteins produce what are regarded as “first order” ciliopathies, but mutations in genes not considered a priori as ciliary (“second-order” ciliopathies) may have a significant influence on ciliary functionality. These genes may be discovered by querying the Human Phenotype Ortology database for phenotypes of established ciliopathies [68]. Several other genes mutated in rare diseases may influence ciliary function. For example, Usher syndrome, the most common cause of genetic deafblindness, is a ciliopathy [69], and patients have been reported to have bronchiectasis [70]. Given the enormous complexity of cilia, it is likely that rare diseases in which the cilia are slightly affected will not be identified as a cause of bronchiectasis.

### 4.3. Potential Connections between Connective Tissue Diseases, Ciliopathies, and Bronchiectasis

Connective tissue diseases (CTD) related to autoimmune diseases, such as rheumatoid arthritis, Sjögren syndrome, or systemic sclerosis, have been viewed as risk factors of bronchiectasis [4] because they are frequently concomitant diseases [71,72,73]. The reasons may be associated, in some cases, with the use of immunosuppressive therapies such as mycophenolate mofetil, believed to induce hypogammaglobulinemia [74]. Another possibility is that the systemic inflammation of the joints in rheumatoid arthritis affects the lungs in some way [75]. The alternative hypothesis that autoimmune diseases are a consequence of bronchiectasis is based on the observation that titers of anti-citrullinated protein antibodies and rheumatoid factor in patients with bronchiectasis and rheumatoid arthritis are increased when compared with those with only rheumatoid arthritis [76]. There is an additional potential way to connect both diseases through respiratory infections, such as staphylococcal and mycobacterial, which induce inflammation leading to bronchiectasis. If the pathogen displays epitopes that mimic self-peptides, an autoimmune disease may develop [77]. Strikingly, abnormal ciliogenesis has been identified in the pathogenesis of autoimmune thyroid diseases [78], linking ciliopathies and autoimmune diseases.

Other CTD such as Marfan or Ehlers–Danlos syndromes, have sometimes been considered aetiologies of bronchiectasis [4,79]. No clear relationship between these syndromes and ciliopathies have been reported, but a mutation in *RIN2* was found in MACS syndrome (Macrocephaly, Alopecia, Cutis laxa, and Scoliosis), an Ehlers–Danlos-like syndrome. In these patients the development of bronchiectasis was described [80]. The component cutis laxa, a heterogeneous group of genetic disorders caused mainly by mutations in the elastin and fibulin-5 genes, has been associated with bronchiectasis in other instances [81]. Scoliosis is also related to PCD [82], and there is a relationship between motile cilia and spine morphology [83]. Additional disorders of the skeletal system, which includes bone and connective tissue, are also produced by mutations in primary cilia genes [84] and are frequently associated with respiratory infections and bronchiectasis. Osteoporosis, which may appear when ciliary genes, such as *IFT80*, *IFT88*, or *Kif3a*, are mutated [84], is considered a risk factor for bronchiectasis [85,86]. Asphyxiating thoracic dystrophy (Jeune’s syndrome) may be caused by mutations in ciliary genes such as *DYNC2H1* [75], and mutations in *DYNC2H1* [87] and in *KIAA0753* [88] were present in patients with recurrent respiratory infections. Mainzer–Saldino syndrome is caused by mutations in *IFT140* [89], but a patient with mutations in the also proposed *TEKT1* presented with lung infections and airway ciliary dyskinesia [90]. Cranioectodermal dysplasia is a consequence of mutations in *WDR35* or *IFT122*. *WDR35* induces the formation of renal cysts [91], but has also been identified in respiratory motile cilia dysfunctions, suggesting an airway mucociliary clearance defect [92]. Patients with *IFT122* mutations are affected by cutis laxa and frequent respiratory infections [93]. The connection between CTD and bronchiectasis has been ingeniously shown by Anne Daniels and collaborators [94]. These researchers analyzed the known frequent enlargement of the dural sac size in CTD, such as Marfan syndrome, and its association with idiopathic bronchiectasis. Marfan syndrome patients displayed the largest dural sac compared to individuals without CTD. Idiopathic bronchiectasis patients also had a significant enlargement of the dural sac, but cystic fibrosis patients did not. The researchers speculated that genetic variants in the TGF- β1 pathway may be partially responsible for the observed results [94], but given the common association between ciliopathies and CTD, these patients may suffer from impaired mucociliary clearance.

### 4.4. Bronchiectasis and Nontuberculous Mycobacteria

The association between ciliopathies and bronchiectasis may explain one of the most common types of mycobacteriosis, Lady Windermere syndrome, caused by *M. avium* complex infection. It shares the nodular/bronchiectatic form of the disease with other mycobacterial infections [95]. Patients are nonsmoker females, with no known predisposing condition, in their sixth decade of life or later. They are thin, often with mitral valve prolapse and have thoracic anomalies, including pectus excavatum and/or scoliosis [95]. It has been earlier described that scoliosis is related to ciliopathies, and pectus excavatum appears regularly in CTD [96], which were also frequently associated with ciliopathies [84]. An interesting result in the above mentioned study involving in the analysis of CTD and idiopathic bronchiectasis [94] was the strong correlation between increased dural sac size and the presence of pulmonary nontuberculous mycobacterial infection. Furthermore, it has been demonstrated that defects in cilia-dependent heart leaflet remodeling during development can prime mitral valve prolapse [49]. Recently, a *MST1R* mutation has been identified as a genetic cause of Lady Windermere syndrome [97]. MST1R is involved in ciliary beat frequency and mucociliary function [98]. However, a relationship between both low weight and gender with ciliopathies is not evident. Nevertheless, it has been proposed that the sex hormone progesterone may play an important role. Progesterone inhibits ciliary beat frequency in the fallopian tube and airway epithelium, and serum concentration is higher and more variable amongst women than men. Serum levels decrease in post-menopausal women to levels registered in men. Therefore, despite reduction in progesterone at advance ages, extensive physiological damage may have occurred in women throughout their lifetimes [99], allowing increased bacterial loads in the lungs. Because all of these conditions related to impaired ciliary function do not appear concurrently in every Lady Windermere syndrome patient, it may be speculated that different genes may be mutated in different patients. Additional evidence about the associations between connective tissue diseases, ciliopathies, and nontuberculous mycobacterial infections is provided by whole exome sequencing from patients with pulmonary nontuberculous mycobacterial disease. A large proportion of these patients have mutations in both cilia and connective tissue genes, much higher than in a control of individuals without the disease [100,101].

Besides Lady Windermere syndrome, the relationship between mycobacteria and bronchiectasis is very close [5]. The controversy about whether mycobacterial infection leads to bronchiectasis or whether bronchiectasis facilitates mycobacterial infections may be unreal, because both events are not mutually exclusive. Nonetheless, the high frequency of bronchiectasis in mycobacteriosis prompts the hypothesis that the mycobacterial infection, following a mucociliary clearance failure, is more frequently previous to bronchiectasis development. The formation of bronchiectasis in GM-CSF knockout mice after experimental *M. abscessus* infection supports the idea that mycobacteria has the capacity to induce an immune response that ends in bronchiectasis [102]. Additionally, computed tomography and histopathological analysis of the evolution of bronchiectasis in mycobacterial infections has prompted several authors to suggest that bronchiectasis was not a pre-existing condition, but resulted from nontuberculous mycobacterial infection [7] (and references therein) as a consequence of a mucociliary clearance failure.

If the hypothesis that mycobacterial infection is the most frequent origin of bronchiectasis is correct, the reasons that explain this fact are unclear. Bacteria, such as *Nocardia* spp., involved in granulomatous diseases, have been reported in patients with bronchiectasias [103]. But it seems that not every microorganism is able to cause bronchiectasis. There are other intracellular bacteria that induce the formation of granuloma, such as *Brucella* spp., which is usually transmitted in humans through the respiratory route. Despite the ensuing inflammatory response, no bronchiectasis developments in infected patients have been described [104]. Other granuloma-forming pathogens are seldom associated with bronchiectasis, including the bacterium *Burkholderia cepacia* [105] and fungi, such as *Histoplasma capsulatum*. However, aspergillosis by *Aspergillus fumigatus* is not rare in bronchiectasis patients [106]. NTM are ubiquitous microorganisms that have adapted to various environmental conditions, including the infection of free-living amoebas, which cause a virulence enhancement. This ability may increase their fitness for intramacrophage survival in human hosts [107]. Among the extracellular bacteria, *Haemophilus influenzae* and *Pseudomonas aeruginosa* are most frequently isolated in patients with bronchiectasis [108], and *P. aeruginosa* has also been proposed to be responsible for bronchiectasis formation [109]. Nevertheless, motile *P. aeruginosa* may move away from static mucus, but immotile mycobacteria may remain for longer periods, allowing multiplication and quorum-sensing-mediated biofilm formation. The interaction between different bacterial species involved in bronchiectasis is complex. The simultaneous isolation of NTM and *P. aeruginosa* is not uncommon [110], and evidence from in vitro studies suggest that *P. aeruginosa* interferes with *M. abscessus* growth [111,112]. In contrast, it has been observed that *P. aeruginosa* growth was promoted in the presence of *M. avium* complex infected cells [113], suggesting that *P.aeruginosa* may benefit from a microenvironment created by a mycobacterial-induced bronchiectasis.

The consequence of the generalized exposure of hosts to environmental mycobacteria may be their normal presence in the lung microbiome. In fact, the role of motile cilia in the airways is not only the removal of particles, but also the creation of fluid-mechanical microhabitats, in which the commensal microbiome will be established [114], allowing a stable permanency of bacteria. Conwan and collaborators sequenced the mycobacterial *hsp65*, a gene usually analyzed for species identification, in DNA extracted from sputum samples. They observed that the gene was detected in all 42 subjects studied, even in the absence of isolation in culture or clinical disease. Furthermore, researchers suggested that NTM are present as communities rather than a single species. Interestingly, the samples from which mycobacteria were not isolated belonged to patients with bronchiectasis [115]. The presence of mycobacteria in the lungs of many, if not all, persons prompts the question about the particular circumstances that led to bronchiectasis development only in a small group of individuals. It is possible that the answer lies in the ability of mycobacteria to grow in biofilms. It may be speculated that mycobacteria has been allowed to assemble biofilm, and will promote bronchiectasis development.

## 5. Hypothetical Course of Bronchiectasis Promoted by NTM and Potential Therapeutical Approaches

From the reviewed data, a plausible description of NTM-mediated bronchiectasis development arises. Many particles, including virus, bacteria, debris, and NTM are inhaled from the environment and trapped in the airway’s mucus. The mucociliary clearance mechanism propels the mucus to the proximal airways and removes them from the lung (Figure 1a). Ciliary movement may be impaired by either primary or secondary dyskinesia (Figure 1C), allowing the mycobacteria to multiply and colonize the bronchial airways as a biofilm or in the form of aggregates (Figure 1b). The immune system responds to the bacterial insult, and leukocytes are attracted to the mycobacterial focus (Figure 1c). A recent in vitro study suggests that this response may be successful because a large enough concentration of neutrophils is able to hinder *M. abscessus* aggregation [116], giving the epithelia the opportunity to restore the normal ciliary function. When the immune response fails, mycobacteria may assemble a biofilm. Fresh neutrophils will continuously replenish the pool to destroy the biofilm through the liberation of numerous enzymes such as proteases that will promote the formation of bronchiectasis (Figure 1d). The newly formed bronchiectasis creates a favorable environment for the colonization of other pathogens, such as *P. aeruginosa*, which will establish their own biofilm and become a chronic infection (Figure 1e).

Current pharmacological therapies for bronchiectasis exacerbations are mainly based on antibiotics. The problem of microbial antibiotic resistance prompts the search for alternative measures. Two potential approaches are the restoration of mucociliary clearance functionality and the impairment of mycobacterial biofilm formation. For ciliary dyskinesia, several strategies are being followed, including the use of eucalypt extracts, anti-inflammatory medications, anti-oxidants, and gene therapy, among others [117]. Pharmacotherapy of ciliopathies may also be useful for ciliary dyskinesia [118]. Of special interest are phosphodiesterase inhibitors. Ciliary function is stimulated by agents that elevate cytoplasmic Ca^2+^, such as cholinergic drugs or adenosine triphosphate (ATP), which can be abolished by phosphodiesterase. A clinical trial has determined that the phosphodiesterase inhibitor sildenafil increases ciliary beat frequency in patients with pulmonary NTM disease [119]. An active search of new drugs taking advantage of new high throughput drug screening methods is underway [120].

Concerns about the importance of bacterial biofilms have prompted much research that deals with their control or elimination. Strategies include anti-biofilm molecules that interfere with bacterial signaling pathways or biofilm dissolving substances. The elimination of the already formed biofilm may be accomplished by dispersal (chemical stimulus that signals bacterial cells to terminate their biofilm lifestyle and exit the exopolymeric substance in which they are embedded) or disruption (degradation of exopolymeric substance) [121]. New strategies are being devised to treat pulmonary biofilm-based chronic infections [122,123]. The poly (acetyl, arginyl) glucosamine is a promising molecule, because both improve mucociliary transport [124], exhibit bactericidal activity, and disperse NTM biofilms [125]. Nevertheless, advances to fight mycobacterial biofilms are hampered by our limited knowledge of mycobacterial quorum sensing and biofilm formation.

## 6. Concluding Remarks

We have presented in this review some of the evidence that is in accordance with the hypothesis that the aetiology of most idiopathic bronchiectasis is a ciliary dysfunction that allows NTM to form a biofilm and initiate the chronic inflammatory response predicted in Cole’s vicious circle model [3]. We propose that the increase in bronchiectasis incidence [126] is not coincident with, but rather a consequence of the increase in mycobacteriosis incidence [127]. In summary, negative results in the diagnosis of PCD does not necessarily mean that cilia are not affected. On the other hand, it is likely that NTM frequently form part of the lung microbiome, but classic microbiologic methods will often fail to isolate mycobacteria from respiratory samples because in some cases of bronchiectasis, NTM infection would be subclinical and not detectable. Therefore, the strict criteria recommended for mycobacterial disease diagnosis by the ATS/IDSA [6] would not be applicable in this context, although it is a recommended guide for patients with clinical mycobacteriosis. Alternative therapeutic measures to antibiotics, such as for restoration of ciliary functionality and against bacterial biofilms, may be applied to incipient bronchiectasis, avoiding its development or even causing its disappearance, as has been observed in cystic fibrosis patients [1].

## Figures and Tables

**Figure 1 microorganisms-12-00665-f001:**
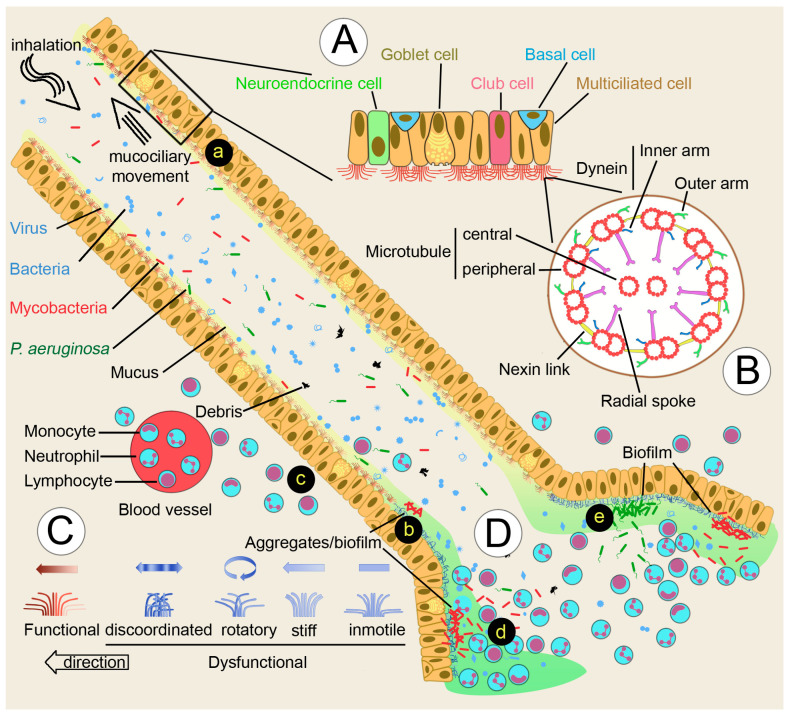
Nontuberculous mycobacteria (NTM) and bronchiectasis. (**A**) Cellular components of multiciliate epithelia. (**B**) Cilium structure. (**C**) Ciliary movement defects occurring in primary or secondary dyskinesia. (**D**) Inflammatory response leading to bronchiectasis. (**a**) Mucociliary clearance of inhaled particles. (**b**) Ciliary dysfunction allows the colonization and growth of mycobacteria in the form of aggregates/biofilm. (**c**) Leukocytes respond to mycobacterial biofilm, but may fail to restrict mycobacterial growth. (**d**) Continuous replenishment of neutrophils, and liberation of neutrophil proteases will promote the formation of bronchiectasis. (**e**) *P. aeruginosa* will colonize and grow as a biofilm, taking advantage of the bronchiectasis microenvironment created by the presence of mycobacteria and the elicited immune response.

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
