# Peer review of "Nontuberculous Mycobacteria, Mucociliary Clearance, and Bronchiectasis"

_microorganisms, 2024, doi:10.3390/microorganisms12040665_

Round 1

Reviewer 1 Report

Comments and Suggestions for Authors

Majo comments:

Indeed, as the author described, bronchiectasis and NTM are closely related, and it is understandable that the number of cases with bronchiectasis is increasing due to the increase in NTM infections. Although various hypotheses exist, neither this narrative review nor previous reports have clarified the mechanisms of NTM infection. In this paper, while the authors focus on cilia dysfunction in their review, mucociliary dysfunction is generally recognized as a critical pathogenic factor contributing to both nontuberculous mycobacteria infections and bronchiectasis. Mucociliary dysfunction, by impairing the clearance of pathogens from the airways, provides a conducive environment for NTM colonization and infection. This common dysfunction for NTM and BE is believed to underlie the mechanisms that significantly increase the incidence of NTM infections and the development of NTM-related bronchiectasis. Furthermore, the current rise in NTM cases is believed to be inexplicable without acknowledging increased interaction with NTM from environment.

Therefore, some of the author's sentences were logically inadequate and need to be corrected, for example “bronchiectasis was not a pre-existing condition but resulted from nontuberculous mycobacterial infection.” and “the high frequency of bronchiectasis in mycobacteriosis suggests that the mycobacterial infection is more frequently previous to bronchiectasis development.”.

Minor comments:

Line231-233

It is true that some PCD patients exhibit normal nasal nitric oxide levels depending on the type of genetic mutation. However, a significant number of patients with PCD exhibit low nasal nitric oxide levels. Therefore, the sentence, “a significant patients with PCD produce normal nitric oxide level”, need to be corrected.

Line264

The author described “the association is not perfect because, for example, no laterality for numerous variants of PCD”. The reason why some cases of PCD do not show situs inversus is related to the fact that the nodal cilia have a 9+0 structure. The frequency of situs inversus even in PCD patients with genetic mutations affecting nodal cilia is 50%.

Please explain a little more about the mechanism of situs inversus in PCD patients so as not to mislead readers.

Line298

As the author described, Marfan and EDS can cause bronchiectasis, and CTD is one of the important aetiologies of bronchiectasis. However, the most common CTD in bronchiectasis patients is rheumatoid arthritis. Therefore, the sentence, “Connective tissue diseases (CTD), such as Marfan or Ehlers-Danlos syndromes, are regularly considered as aetiologies of bronchiectasis.”, need to be corrected. The author described RA and Sjogren syndrome in the following paragraph. However, please consider rearranging the order of the paragraphs.

Line332

The mechanism of bronchiectasis caused by rheumatoid arthritis involves not only the effects of immunosuppressants but also airway inflammation. Furthermore, RA patients are more susceptible to infection. Please explain a little more about the mechanisms of bronchiectasis caused by RA.

Line381-389

It is good that you mentioned the various bacteria. In recent years, it has been reported that Nocardia is closely related to bronchiectasis. Can you add an explanation of Nocardia?

Line393

Haemophilus influenzae is more appropriate than Haemophilus influenza.

Line 475

As the author described, sometimes, NTM cannot be detected by sputum culture. However, the presence of NTM in the microbiome may not have clinical significance; NTM is a bacterium that frequently causes contamination. Therefore, adherence to ATS/IDSA is important in clinical situations.

Author Response

Response to reviewer 1

Reviewer 1

Indeed, as the author described, bronchiectasis and NTM are closely related, and it is understandable that the number of cases with bronchiectasis is increasing due to the increase in NTM infections. Although various hypotheses exist, neither this narrative review nor previous reports have clarified the mechanisms of NTM infection. In this paper, while the authors focus on cilia dysfunction in their review, mucociliary dysfunction is generally recognized as a critical pathogenic factor contributing to both nontuberculous mycobacteria infections and bronchiectasis. Mucociliary dysfunction, by impairing the clearance of pathogens from the airways, provides a conducive environment for NTM colonization and infection. This common dysfunction for NTM and BE is believed to underlie the mechanisms that significantly increase the incidence of NTM infections and the development of NTM-related bronchiectasis. Furthermore, the current rise in NTM cases is believed to be inexplicable without acknowledging increased interaction with NTM from environment.

  Therefore, some of the author's sentences were logically inadequate and need to be corrected, for example “bronchiectasis was not a pre-existing condition but resulted from nontuberculous mycobacterial infection.” and “the high frequency of bronchiectasis in mycobacteriosis suggests that the mycobacterial infection is more frequently previous to bronchiectasis development.”.

 We agree with the reviewer, and that is our hypothesis. Nevertheless, it seems that we did not explain it clearly. First, there is a mucociliary clearance failure, then infection that leads to bronchiectasis, and what we suggest is that infection is more frequently by nontuberculous mycobacteria than currently acknowledged. We changed the first sentence as follows (line 402):

Additionally, computed tomography and histopathological analysis of the evolution of bronchiectasis in mycobacterial infections has prompted several authors to suggest that bronchiectasis was not a pre-existing condition but resulted from nontuberculous mycobacterial infection [7] (and references therein) as a consequence of a mucociliary clearance failure.

The second sentence was changed as follows (line 394):

Nonetheless, the high frequency of bronchiectasis in mycobacteriosis suggests that the mycobacterial infection, following a mucociliary clearance failure, is more frequently previous to bronchiectasis development.

Minor comments:

Line231-233

It is true that some PCD patients exhibit normal nasal nitric oxide levels depending on the type of genetic mutation. However, a significant number of patients with PCD exhibit low nasal nitric oxide levels. Therefore, the sentence, “a significant patients with PCD produce normal nitric oxide level”, need to be corrected.

 We have changed the sentence as follows (line 245)

some patients with PCD produce normal nitric oxide levels

Line264

The author described “the association is not perfect because, for example, no laterality for numerous variants of PCD”. The reason why some cases of PCD do not show situs inversus is related to the fact that the nodal cilia have a 9+0 structure. The frequency of situs inversus even in PCD patients with genetic mutations affecting nodal cilia is 50%.

Please explain a little more about the mechanism of situs inversus in PCD patients so as not to mislead readers.

 Agree. We have added an extra explanation as follows (line 277):

The reason is that some genes characteristic of 9+2 cilia mutated in PCD, such as those encoding proteins of the radial spoke (Figure 1.B), are not expressed in the 9+0 cilia responsible for laterality.

Line298

As the author described, Marfan and EDS can cause bronchiectasis, and CTD is one of the important aetiologies of bronchiectasis. However, the most common CTD in bronchiectasis patients is rheumatoid arthritis. Therefore, the sentence, “Connective tissue diseases (CTD), such as Marfan or Ehlers-Danlos syndromes, are regularly considered as aetiologies of bronchiectasis.”, need to be corrected. The author described RA and Sjogren syndrome in the following paragraph. However, please consider rearranging the order of the paragraphs.

 Agree. We have changed the order of the paragraphs. Comments regarding to autoimmune diseases are placed first. We have changed the sentence as follows (line328):

Other CTD such as Marfan or Ehlers-Danlos syndromes, are regularly have sometimes been considered as aetiologies of bronchiectasis

Line332

The mechanism of bronchiectasis caused by rheumatoid arthritis involves not only the effects of immunosuppressants but also airway inflammation. Furthermore, RA patients are more susceptible to infection. Please explain a little more about the mechanisms of bronchiectasis caused by RA.

 Agree. We have included a new reference in which is suggested the participation of systemic inflammation as follows (line 317):

Another possibility is that the systemic inflammation of the joints in rheumatoid arthritis somehow also affects the lungs

Line381-389

It is good that you mentioned the various bacteria. In recent years, it has been reported that Nocardia is closely related to bronchiectasis. Can you add an explanation of Nocardia?

 Agree. A reference has been added regarding Nocardia, as follows (line 404):

Bacteria, such as Nocardia spp., involved in granulomatous diseases, have been reported in patients with bronchiectasias.

Line393

Haemophilus influenzae is more appropriate than Haemophilus influenza.

 Corrected. Thank you for your comment.

Line 475

As the author described, sometimes, NTM cannot be detected by sputum culture. However, the presence of NTM in the microbiome may not have clinical significance; NTM is a bacterium that frequently causes contamination. Therefore, adherence to ATS/IDSA is important in clinical situations.

Agree. To stress this point, we have added the following sentence (line 500):

although it is a recommended guide for patients with clinical micobacteriosis.

Reviewer 2 Report

Comments and Suggestions for Authors

Dear Authors,

I read manuscript entitled: Nontuberculous mycobacteria, mucociliary clearance and bronchiectasis

In this manuscript, the authors propose that ciliary dyskinesia is the main promoter of bronchiectasis and that the bacteria most frequently involved are NTMs. Consequently, the restoration of ciliary function and the reduction of mycobacterial biofilm formation are suggested as potential effective therapeutic alternatives to antibiotics.

The topic of the review is very interesting, but the paper lacks proper organization and thorough description. It becomes excessively detailed in the anatomical, physiological, and pathological aspects of well-known pathologies, lacking completeness in describing the pathological mechanism that associates bronchiectasis with ciliary dysfunction and biofilm formation of nontuberculous mycobacteria. In addition, not all cited articles have been accurately interpreted.

I have some concerns to raise that bear mentioning:

1) Regarding the ‘Introduction’:

§  The statement "Bronchiectasis are dilatations of the bronchi, no longer considered always irreversible" contradicts established literature. According to the British Thoracic Society Guideline for bronchiectasis in adults (reference: Thorax. 2019 Jan; doi: 10.1136/thoraxjnl-2018-212463), bronchiectasis is indeed irreversible.

The cited article [1] in the manuscript suggests a need for robust data regarding the potential reversibility of bronchiectasis in adults.

2) In the section on ‘Nontuberculous mycobacteria’:

§  This chapter is overly verbose and does not accurately describe the main topic: mycobacterial biofilms and their association with bronchiectasis.

3) In the 'Airway Mucociliary Clearance' section:

§  This chapter primarily comprises a description of the anatomy and physiology of the respiratory system, exhibiting excessive verbosity.

4) In the section on 'Bronchiectasis and Ciliopathies':

§  The chapter title is misleading. The text does not describe the association between ciliopathies and bronchiectasis, but rather outlines the clinical manifestations associated with ciliopathies and bronchiectasis.

5) Regarding the ‘Bronchiectasis and connective tissue diseases’:

§  In this chapter, the focus extends beyond the association between bronchiectasis and connective tissue diseases, as implied by the title. It also delves into the potential connections between bronchiectasis, ciliopathies, and connective tissue diseases.

§  The chapter excessively describes the numerous clinical manifestations associated with ciliopathies, losing sight of the main topic.

6) In reference to the 'Bronchiectasis and nontuberculous mycobacteria' section::

§  The citation [28] pertains to a small group of patients with cystic fibrosis and the M. abscessus complex, rather than covering all mycobacteria and every patient as described in the manuscript: ‘The presence of mycobacteria in the lungs of many, if not all, persons prompts the question about the particular circumstances that lead to bronchiectasis development only in a small group of individuals. It is possible that the answer lies in the ability of mycobacteria to grow in biofilms [28]’.

It would be advisable to include more literature references to support the pathological mechanism underlying  your hypothesis ( aetiology of most of the idiopathic bronchiectasis is a ciliary dysfunction that allows NTM to form biofilm and initiate the chronic inflammatory response predicted in Cole's vicious circle model).

Author Response

Response to reviewer 2

Reviewer 2

Dear Authors,

I read manuscript entitled: Nontuberculous mycobacteria, mucociliary clearance and bronchiectasis

In this manuscript, the authors propose that ciliary dyskinesia is the main promoter of bronchiectasis and that the bacteria most frequently involved are NTMs. Consequently, the restoration of ciliary function and the reduction of mycobacterial biofilm formation are suggested as potential effective therapeutic alternatives to antibiotics.

The topic of the review is very interesting, but the paper lacks proper organization and thorough description. It becomes excessively detailed in the anatomical, physiological, and pathological aspects of well-known pathologies, lacking completeness in describing the pathological mechanism that associates bronchiectasis with ciliary dysfunction and biofilm formation of nontuberculous mycobacteria. In addition, not all cited articles have been accurately interpreted.

I have some concerns to raise that bear mentioning:

1) Regarding the ‘Introduction’:

  • The statement "Bronchiectasis are dilatations of the bronchi, no longer considered always irreversible" contradicts established literature. According to the British Thoracic Society Guideline for bronchiectasis in adults (reference: Thorax. 2019 Jan; doi: 10.1136/thoraxjnl-2018-212463), bronchiectasis is indeed irreversible.

The cited article [1] in the manuscript suggests a need for robust data regarding the potential reversibility of bronchiectasis in adults.

Agree. The reviewer is right when saying that it is not established that bronchiectasis may be reversible. We have changed the sentence and only indicate that it is speculated the possibility of reversibility (line 39):

Currently regarded as irreversible, some reports have lately speculated about potential improvements after chemotherapy

2) In the section on ‘Nontuberculous mycobacteria’:

  • This chapter is overly verbose and does not accurately describe the main topic: mycobacterial biofilms and their association with bronchiectasis.

Agree. We have removed some text that was not necessary to explain our hypotheses. We have included additional information regarding biofilm development in mycobacteria. There are no reports in which are shown biofilms in bronchiectasis. We comment this point as follows (line 120):

Although the development of mycobacterial biofilm is expected in bronchiectasis, no reports have been published, probably due to a scarcity of studies.

3) In the 'Airway Mucociliary Clearance' section:

  • This chapter primarily comprises a description of the anatomy and physiology of the respiratory system, exhibiting excessive verbosity.

Agree. We have removed some text that was not necessary to explain our hypotheses.

4) In the section on 'Bronchiectasis and Ciliopathies':

  • The chapter title is misleading. The text does not describe the association between ciliopathies and bronchiectasis, but rather outlines the clinical manifestations associated with ciliopathies and bronchiectasis.

Agree. We have changed the title to Concurrent ciliopathies and bronchiectasis (line 252). We hope that the various examples of their simultaneous clinical manifestations help to support the hypothesis that they are mechanistically related.

5) Regarding the ‘Bronchiectasis and connective tissue diseases’:

  • In this chapter, the focus extends beyond the association between bronchiectasis and connective tissue diseases, as implied by the title. It also delves into the potential connections between bronchiectasis, ciliopathies, and connective tissue diseases.

Agree. We have changed the title to Potential connections between connective tissue diseases, ciliopathies and bronchiectasis (line 511).

  • The chapter excessively describes the numerous clinical manifestations associated with ciliopathies, losing sight of the main topic.

The reason for including so many diseases in which simultaneously appear connective tissue diseases, confirmed or potential ciliopathies, and bronchiectasias or respiratory infections is that we have tried to be comprehensive. We suppose that many other researchers have realized that those connections exist. But we are not aware of any original article or review that detail them. We think that it is one of the strong points of the review, because it is a novel approach, and it is important for our line of reasoning. The main example is Lady Windermere syndrome, explained in the following section (Bronchiectasis and nontuberculous mycobacteria). We have provided many evidences that this syndrome is very likely a ciliopathy. In fact, this syndrome is the main reason why we have focused in ciliopathies related to connective tissue diseases.

6) In reference to the 'Bronchiectasis and nontuberculous mycobacteria' section::

  • The citation [28] pertains to a small group of patients with cystic fibrosis and the M. abscessus complex, rather than covering all mycobacteria and every patient as described in the manuscript: ‘The presence of mycobacteria in the lungs of many, if not all, persons prompts the question about the particular circumstances that lead to bronchiectasis development only in a small group of individuals. It is possible that the answer lies in the ability of mycobacteria to grow in biofilms [28]’.

Agree. We have removed the reference from this paragraph.

It would be advisable to include more literature references to support the pathological mechanism underlying  your hypothesis ( aetiology of most of the idiopathic bronchiectasis is a ciliary dysfunction that allows NTM to form biofilm and initiate the chronic inflammatory response predicted in Cole's vicious circle model).

Agree. We think that studies based in whole genome sequencing of patients infected with nontuberculous mycobacteria provide interesting clues about the importance of mutations in cilia and connective tissue diseases. We have added this information in line 384 as follows:

Additional evidence about the associations between connective tissue diseases, ciliopathies and nontuberculous mycobacterial infections is provided by whole exome sequencing from patients from pulmonary nontuberculous mycobacterial disease. A large proportion of these patients have mutations in both cilia and connective tissue genes, much higher than in a control of individuals without the disease.

Round 2

Reviewer 1 Report

Comments and Suggestions for Authors

The revised manuscript has been improved according to the suggestions by reviewers. However, there are several points that still need to be changed.

  • Line 315-317 The text currently suggests that immunosuppressants are the primary cause of bronchiectasis, which needs correction. Line 317-318 The sentence 'Arthritis affects the lungs in some way' needs to be revised. Line 393-395 It should be more strongly emphasized that this is a hypothesis.

Author Response

The revised manuscript has been improved according to the suggestions by reviewers. However, there are several points that still need to be changed.

  • Line 315-317 The text currently suggests that immunosuppressants are the primary cause of bronchiectasis, which needs correction. Line 317-318 The sentence 'Arthritis affects the lungs in some way' needs to be revised. Line 393-395 It should be more strongly emphasized that this is a hypothesis.

We thank the reviewer for the continuous effort to enhance our manuscript. All suggestions have strongly improved the readability and accuracy of the presented ideas. We have changed the sentences, following the reviewer’s comments (highlighted in pink in the original text), as follows:

- Previous text (lines 315-317 and 317-319)

The reasons may be associated to the use of immunosuppressive therapies such as mycophenolate mofetil, believed to induce hypogammaglobulinemia [76]. Another possibility is that the systemic inflammation of the joints in rheumatoid arthritis somehow also affects the lungs [77].

Corrected

The reasons may be associated, in some cases, to the use of immunosuppressive therapies such as mycophenolate mofetil, believed to induce hypogammaglobulinemia [76]. Another possibility is that the systemic inflammation of the joints in rheumatoid arthritis affects the lungs in some way [77].

- Previous text (lines 393-395)

The controversy about whether mycobacterial infection leads to bronchiectasis or bronchiectasis facilitates mycobacterial infections may be unreal, because both events are not mutually exclusive. Nonetheless, the high frequency of bronchiectasis in mycobacteriosis suggests that the mycobacterial infection, following a mucociliary clearance failure, is more frequently previous to bronchiectasis development.

Corrected

The controversy about whether mycobacterial infection leads to bronchiectasis or bronchiectasis facilitates mycobacterial infections may be unreal, because both events are not mutually exclusive. Nonetheless, the high frequency of bronchiectasis in mycobacteriosis prompts the hypothesis that the mycobacterial infection, following a mucociliary clearance failure, is more frequently previous to bronchiectasis development.